# How enzymatic activity is involved in chromatin organization

**Rakesh Das[1]\*, Takahiro Sakaue[2], GV Shivashankar[3,4], Jacques Prost[1,5], Tetsuya Hiraiwa[1]\***

[1]Mechanobiology Institute, National University of Singapore, Singapore, Singapore; [2]Department of Physics and Mathematics, Aoyama Gakuin University, Kanagawa, Japan; [3]ETH Zurich, Zurich, Switzerland; [4]Paul Scherrer Institute, Villigen, Switzerland; [5]Laboratoire Physico Chimie Curie, Institut Curie, Paris Science et Lettres Research University, Paris, France

**Abstract** Spatial organization of chromatin plays a critical role in genome regulation. Previously, various types of affinity mediators and enzymes have been attributed to regulate spatial organization of chromatin from a thermodynamics perspective. However, at the mechanistic level, enzymes act in their unique ways and perturb the chromatin. Here, we construct a polymer physics model following the mechanistic scheme of Topoisomerase-II, an enzyme resolving topological constraints of chromatin, and investigate how it affects interphase chromatin organization. Our computer simulations demonstrate Topoisomerase-II's ability to phase separate chromatin into eu- and heterochromatic regions with a characteristic wall-like organization of the euchromatic regions. We realized that the ability of the euchromatic regions to cross each other due to enzymatic activity of Topoisomerase-II induces this phase separation. This realization is based on the physical fact that partial absence of self-avoiding interaction can induce phase separation of a system into its self-avoiding and non-self-avoiding parts, which we reveal using a mean-field argument. Furthermore, motivated from recent experimental observations, we extend our model to a bidisperse setting and show that the characteristic features of the enzymatic activity-driven phase separation survive there. The existence of these robust characteristic features, even under the non-localized action of the enzyme, highlights the critical role of enzymatic activity in chromatin organization.

**\*For correspondence:**
rakeshd68@yahoo.com (RD);
mbithi@nus.edu.sg (TH)

**Competing interest:** The authors declare that no competing interests exist.

## Editor's evaluation

This manuscript will be of interest to readers in the field of physical biology and molecular biology for understanding genome organization. The idea of this computational study and its outcomes suggest a novel phase-separated structure and will shed new light on the role of enzymatic activity in chromatin organization. Overall, modeling and simulation are properly performed and analyzed, and the data support the key claims of the manuscript.

## Introduction

During interphase, chromatin in a nucleus is densely packed and unable to move freely around the nucleus, resulting in a highly constrained positioning of genes. Nowadays, it is acknowledged that such physical spacing of chromatin (genes) is critical in regulating biochemical and transcriptional abilities of genes (*Uhler and Shivashankar, 2017*; *Wang et al., 2018*; *Elgin and Reuter, 2013*), and proper functionality of the genomic content depends on the nonrandom organization of chromatin (*Solovei et al., 2016*; *Hildebrand and Dekker, 2020*). Three-dimensional contact mapping techniques have revealed that chromatin is compartmentalized into euchromatic (EC) and heterochromatic (HC)

regions (*Lieberman-Aiden et al., 2009*; *Fiorillo et al., 2021*). In the EC regions, the nucleosomes are widely separated allowing greater access of the embedded genes to various regulatory factors, and therefore, EC regions are transcriptionally active. In contrast, HC regions comprise densely packed nucleosomes, and they are transcriptionally repressed. Recent literature *Larson et al., 2017*; *Larson and Narlikar, 2018*; *Strom et al., 2017*; *Gibson et al., 2019*; *Erdel and Rippe, 2018*; *Hildebrand and Dekker, 2020* have argued phase separation as one of the driving mechanisms for such compartmentalization of chromatin. Affinity among HC regions, mediated by a diverse range of molecular agents (*Erdel and Rippe, 2018*; *Hildebrand and Dekker, 2020*), is believed to induce such phase separation in chromatin. Besides this affinity-induced phase separation, many active agents (which are ATP dependent and therefore capable of driving the system out of equilibrium) play crucial roles in chromatin organization, for example, extruder-motor assisted loop formation (*Nuebler et al., 2018*; *Mirny et al., 2019*) or RNA polymerase II mediated transcriptional pocket formation (*Hilbert et al., 2021*).

Nuclear media is full of various types of affinity mediators and active agents. To investigate how those agents affect chromatin organization, it can be useful to employ concepts of physics. As a matter of fact, polymer physics models have been successfully employed to explain various aspects of experimental observations (*Lieberman-Aiden et al., 2009*; *Imakaev et al., 2015*; *Fiorillo et al., 2021*). Modeling chromatin as block copolymers and tuning the affinity among those blocks could reproduce the plaid-like pattern observed in contact maps (*Jost et al., 2014*; *Falk et al., 2019*; *MacPherson et al., 2018*). Here, the blocks represent genomic regions with different epigenetic marks, for example, H3K9ac and H3K27me3 histone marks characterizing EC and HC regions, respectively. Polymer physics approach has also been useful to implicate the role of active biophysical processes on chromatin organization (*Smrek and Kremer, 2017*; *Ganai et al., 2014*; *Agrawal et al., 2020*). By modeling active sites of active agents as local regions at higher temperatures as compared to the rest of the media, these studies highlighted the effect of out-of-equilibrium processes on chromatin organization. However, at the mechanistic level it is likely that the activity of each enzyme will affect dynamics beyond just effective-temperature inhomogeneity. We need dedicated studies to elucidate how the enzymatic activity can affect the microphase separation (MPS) structures beyond just a thermodynamics phenomenology by employing the mechanistic model focusing on a specific type of enzyme.

In this paper, we focus on topoisomerase enzyme of type II (Topo-II), an active agent that plays a pivotal role in resolving topological constraints of chromatin which emerge due to dense packing (*Nitiss, 2009*; *Vologodskii, 2016*; *Roca, 2009*; *Pommier et al., 2016*; *Baranello et al., 2013*; *Chen et al., 2013*; *Poljak and Käs, 1995*), and investigate the effects of this enzyme on chromatin organization. Topo-II transports one DNA duplex across another, which is cleaved transiently and resealed after transport. The role of this enzyme in processes like transcription, replication, and segregation of sister chromatids has been investigated extensively (*Nitiss, 2009*; *Pommier et al., 2016*; *Ju et al., 2006*); here, we investigate the possibility for this enzyme to modify chromatin organization during interphase. To accomplish this aim, we developed an active polymer model mimicking the mechanistic scheme of Topo-II's activity. We find that Topo-II has inherent ability to induce MPS in chromatin. Using simplified model studies, we argue that the underlying mechanism of Topo-II-driven phase separation is of a new type; the effective phantomness of polymer segments (i.e., the ability of the segments crossing each other) due to Topo-II activity induces phase separation. We find that Topo-II induces a characteristic 'wall'-like structure of EC regions – a feature that has not been observed in other models studying phase separation of chromatin. Further, we investigate how such MPS structure is affected by bidispersity of the chromatin. The idea of considering the case of a bidisperse chromatin is inspired from *Xu et al., 2018*, which showed that epigenetic marks associated with EC and HC regions remains as clusters of *different sizes*.

## Results

### Polymer model of Topoisomerase's activity

We developed a copolymer model to study three-dimensional organization of a 50.807 Mbp long chromatin confined within a spherical cavity of diameter 1.4112 μm. The copolymer comprises two types of equal-sized beads, A and B, connected by springs (*Figure 1a*). These beads represent EC

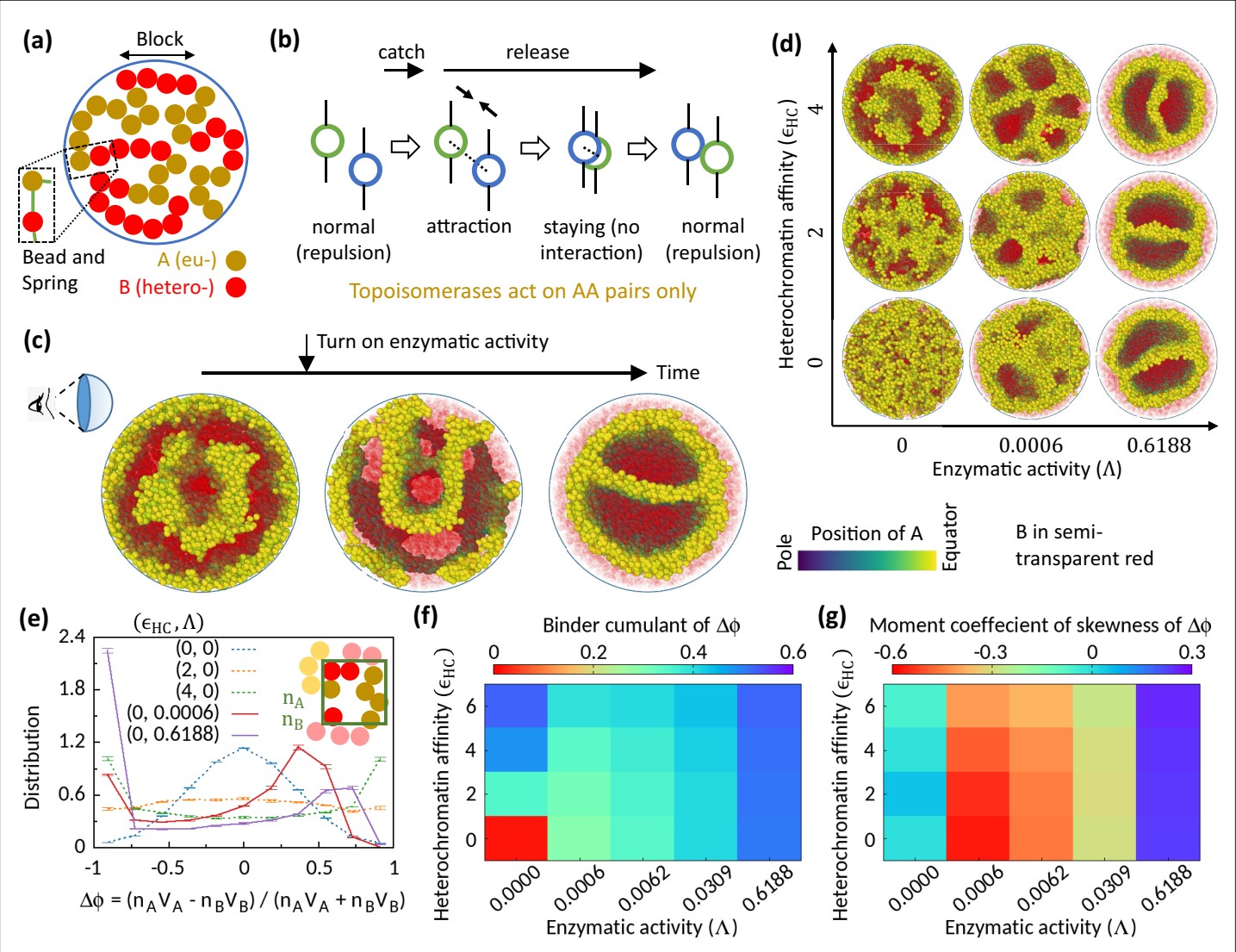

**Figure 1.** Microphase separation of eu- and heterochromatic regions due to enzymatic activity. (**a**) A random multiblock copolymer comprising A and B beads connected by springs confined within a spherical cavity. All the data are shown for block size $b = 4$. (**b**) Topo-II enzyme catches two A's in spatial neighborhood. Through a sequence of processes, it passes one A across another with some probability and eventually releases both A's. (**c**) Sample snapshots (hemisphere cuts) showing that microphase separation configuration changes significantly after turning on enzymatic activity. The color bar indicates position of A's, and B's are shown in semi-transparent red. Parameters used—$\epsilon_{HC} = 4$ and $\Lambda = 0.6188$. (**d**) Sample snapshots showing microphase separation in response to heterochromatin affinity and enzymatic activity. (**e**) Inset—The cavity is divided into small grids, and $n_A$ and $n_B$ stand for the numbers of the respective beads within individual grid. Main—$V_{A,B}$ represent volume of the respective beads. Distribution of $\Delta\phi$ goes from unimodal to bimodal as the system phase separates. Time-averaged data shown, and error bars indicate standard deviations over four realizations. (**f**) Binder cumulant $1 - \langle(\Delta\phi)^4\rangle_P/3\langle(\Delta\phi)^2\rangle_P^2$ value greater than zero indicates deviation of the $\Delta\phi$-distribution from the Gaussian profile. (**g**) Moment coefficient of skewness $E\left[(\Delta\phi - \langle\Delta\phi\rangle)^3\right] / \left\{E\left[(\Delta\phi - \langle\Delta\phi\rangle)^2\right]\right\}^{3/2}$ captures the asymmetry in the $\Delta\phi$-distribution about its mean in the presence of Topo-II.

and HC contents, respectively, each mirroring 2.4203 kbp of chromatin (see Methods). Each type of bead appears in blocks of size $b$, and the blocks are distributed randomly along the polymer. At any instant of time, an individual bead may realize potential energy fields due to (i) its connectivity to its neighbors along the polymer ($H_{spring}$ in *Equation 2*; see Methods), (ii) its finite volume resulting a steric repulsion ($h_{vex} > 0$), (iii) affinity among HC regions ($H_{HC}$), and (iv) confinement by the cavity-boundary ($H_{confinement}$). Affinity among HC regions is modeled by a short range attraction between B beads, parameterized by $\epsilon_{HC}$ (*Erdel and Rippe, 2018*; *Hildebrand and Dekker, 2020*; *Falk et al., 2019*; *Nuebler et al., 2018*).

Topo-II relaxes topological constraints of a chromatin in a *catch-and-release* mechanism—it catches two DNA segments in spatial proximity, and through a sequence of processes including ATP hydrolysis, it eventually transports one DNA segment across the other and releases both segments (*Roca, 2009*; *Nitiss, 2009*). We engineered our polymer model in a particular way to mimic this catch-and-release mechanism of Topo-II's activity. First, a Topo-II catches two beads in spatial proximity (within unit length separation in simulation units) with a Poisson rate $\lambda_{ra}$ (*Figure 1b*). The beads bound to the enzyme no more exert steric repulsion to each other; instead they attract each other (i.e., $h_{vex} < 0$). This attraction state mimics the locked N-gate state of Topo-II that brings two DNA segments closer to each other (*Roca, 2009*). Next, the attraction between those two beads is turned off with a rate $\lambda_{an}$, and the beads stay there for a while without any steric interaction among themselves (i.e., $h_{vex} = 0$). This step allows the beads to pass across each other stochastically. Eventually, the enzyme unbind from the beads with a rate $\lambda_{nr}$, and the beads return to their normal state with steric repulsion between themselves. We assume that the rates are uniform across the cavity. These rates statistically determine the times which a proximal pair of beads spend in the steric repulsion state, attraction state, or no interaction state. We define enzymatic activity as $\Lambda = \lambda_{ra} \left(1/\lambda_{an} + 1/\lambda_{nr}\right)$, which can be tuned in experiments by controlling ATP concentration (*Lindsley and Wang, 1993*).

Experiments using budding yeast suggests that Topo-II mainly works on the nucleosome-free regions of the genome (*Sperling et al., 2011*; *Baranello et al., 2013*). As it is more likely for Topo-II to find nucleosome-free bare DNA segments in the EC regions, we assume that Topo-II works on AA pairs only. Also, we assume that the two beads caught by a Topo-II are not immediate neighbors along the polymer, as this is less likely to be the case. As we mention later, we have checked that this assumption does not qualitatively affect the results presented below. Hereafter, we refer to this polymer model as the *monodisperse differential active model* (MdDAM). We simulate this model using Brownian dynamics at physiological temperature 310 K (see Methods). The composition of the copolymer system is quantified by volume fraction $\phi_A$ of A beads, defined as the ratio of the total volume of A beads to that of all the beads.

## Topoisomerase affects chromatin organization

To investigate the role of Topo-II on chromatin organization, we compare the morphology of chromatin organization in the absence and the presence of enzymatic activity. We start our simulation in the absence of Topo-II but for finite HC affinity ($\epsilon_{HC} > 2$) and observe MPS of chromatin into EC-rich and HC-rich domains (*Figure 1c*—left). Interestingly, once the enzymatic activity is turned on during the simulation, the MPS structure evolves into a significantly different morphology (*Figure 1c*—center and right). This suggests the importance of Topo-II's activity in chromatin organization.

Next, we focus on the differences between the MPS induced by HC affinity and that due to Topo-II's activity. In the absence of Topo-II, the chromatin microphase separates in response to HC affinity, as shown in *Figure 1d* for symmetric composition ($\phi_A = 0.5$). To quantify this MPS, we define an order parameter $\Delta\phi = \left(n_A V_A - n_B V_B\right)/\left(n_A V_A + n_B V_B\right)$, where $V_{A,B}$ represents the volume of the respective beads, and $n_{A,B}$ represents the number of the corresponding beads in a coarse-grained locality. The system remains in a disordered state in the control case ($\epsilon_{HC} = 0$), and the distribution $P\left(\Delta\phi\right)$ (see Methods) of the order parameter follows a Gaussian curve (*Figure 1e*). However, $P\left(\Delta\phi\right)$ flattens and eventually becomes bimodal with HC affinity, suggesting the appearance of EC-rich and HC-rich domains. This phenomenology is well captured in the phase diagram shown in *Figure 1f*, where the Binder cumulant increases with $\epsilon_{HC}$ suggesting deviation of $P\left(\Delta\phi\right)$ from the Gaussian distribution.

Interestingly, Topo-II not only alters chromatin organization, but alone can drive MPS of chromatin (*Figure 1d*). In the presence of Topo-II, $P\left(\Delta\phi\right)$ becomes bimodal (*Figure 1e*), which signifies Topo-II's ability to induce MPS. However, the bimodal profile of $P\left(\Delta\phi\right)$ is strongly asymmetric about its mean in the presence of enzymatic activity, as shown in the distribution (*Figure 1e*) and quantified by skewness (*Figure 1g*). This is the stark difference from the affinity-induced case, where we have a symmetric profile of order parameter distribution around $\Delta\phi = 0$ (*Figure 1e and g*), as expected for the symmetric composition. This suggests that the phase separation induced by the Topo-II activity attributes to a fundamentally new mechanism that is qualitatively distinct from the affinity-induced case.

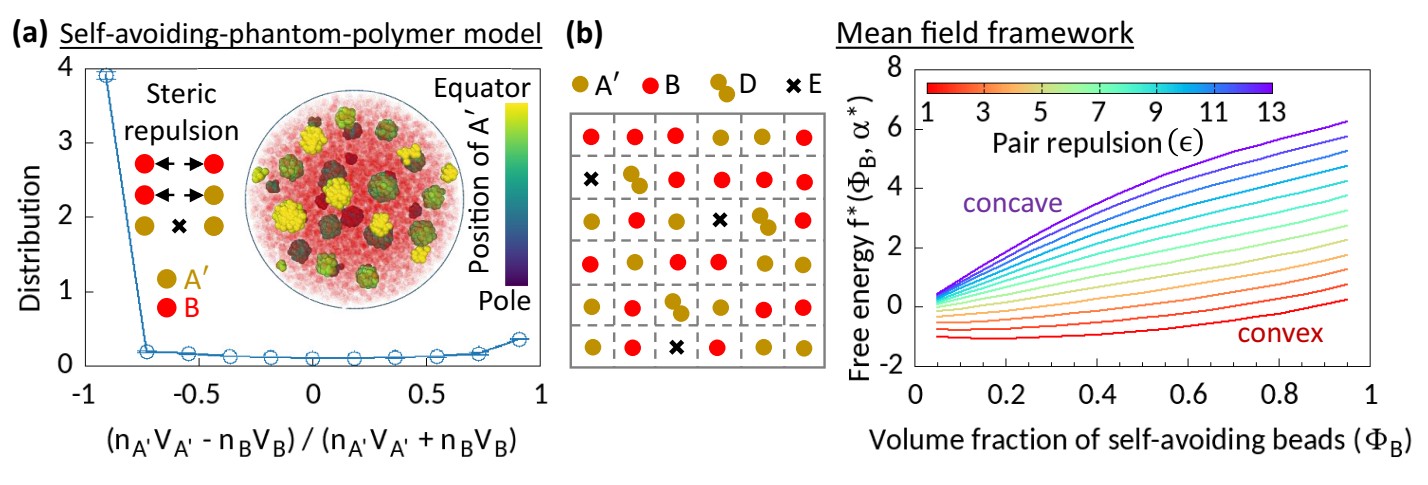

**Figure 2.** Phase separation in system comprising self-avoiding and phantom regions. (**a**) An equilibrium copolymer system comprising phantom (A') and self-avoiding (B) beads is simulated in the absence of steric interaction between A''s. The system shows microphase separation. A sample snapshot (hemisphere cut) is shown where B's are in semi-transparent red. Time-averaged data shown for the distribution, and error bars indicate standard deviations over five realizations. (**b**) Left—Schematic of a lattice space filled with phantom (A') and self-avoiding (B) beads. A' beads can form doublets (D) resulting empty (E) sites. Mean-field calculation gives an effective attraction among B's. Right—Free energy $f^*$ curves drawn for critical doublet fraction $\alpha^*(\Phi_B, \epsilon)$ shows convex to concave transition with pair repulsion parameter $\epsilon$, suggesting a phase separation in the system.

## Plausible mechanism of enzymatic activity-driven phase separation

To understand the underlying physical mechanism of enzymatic activity-driven phase separation, we reduce MdDAM to its equilibrium version. We replace the self-avoiding A beads of MdDAM by phantom A' beads such that there is no steric interaction between A'A' pairs at any time. Interestingly, this simplified model, called a self-avoiding-phantom-polymer model, shows MPS even in the absence of HC affinity (*Figure 2a*). A''s prefer other A''s as their neighbors because that saves the steric energy cost of the system. Moreover, the number of available microstates and therefore the entropy of the system increases if A''s stay close to each other. We argue that both these energetic and entropic advantages drive MPS in this equilibrium polymer system. From a physics standpoint, it would be interesting to construct a mean-field framework for the self-avoiding-phantom-polymer model in the spirit of *Fredrickson et al., 1992*; however, that is a non-trivial task and beyond the scope of the current article. Instead, we construct a relevant but reduced mean-field framework by relaxing the polymeric degrees of freedom of the system. In this simplification, the MPS morphology observed for the self-avoiding-phantom-polymer model will vanish, but the fundamental mechanism driving the phase separation should still be at work. We discuss this simplified mean-field framework in the next paragraph.

We consider a lattice system filled with A' and B beads (*Figure 2b*). Two A' beads can form a doublet (D), resulting in an empty lattice site (E). Given a doublet fraction $\alpha \in [0, 1/2]$, our mean-field level calculation gives a term $\epsilon c(\alpha)\Phi_B^2$ in the free energy density, where $\Phi_B$ is the volume fraction of B's in this mean-field model, and the coefficient $c(\alpha) < 0$ (see Methods). This term suggests that the phantomness of A' eventually induces an effective attraction between B's, driving a phase separation in the system. The observed convex to concave transition of the profile of the reduced free energy density $f^*$ expressed for the critical doublet fraction $\alpha^*$, with pair repulsion parameter $\epsilon$ justifies our claim (*Figure 2b*).

A's in MdDAM transiently behave like phantom beads. Seeing how phantomness of one type of beads can induce a phase separation in the simplified model systems described above, we argue that a similar physical mechanism is responsible for the enzymatic activity-driven phase separation. However, we must note that the wall-like organization of A beads in the Topo-II-driven microphase separated configurations, as elaborated in the next section, is a unique feature not observed in the phase separation between self-avoiding and phantom segments (*Figure 2a*).

In the snapshots shown for $\Lambda = 0.6188$ in *Figure 1d*, the phase separation structure is not affected by the HC affinity. Probably, the energy cost derived from the entropic advantage due to Topo-II's

activity might overpower HC affinity, resulting in this robust phase separation structure for strong enzymatic activity.

## Topo-II induces characteristic phase separation features, including wall-like organization of EC

We examine the obtained microphase separated snapshots to understand the effect of Topo-II on chromatin organization. The number density of the beads suggests an alternating and complimentary organization of A and B beads along the radius of the cavity (*Figure 3a*). Interestingly, in the presence of enzymatic activity, we note a wall-like appearance of EC domains (*Figure 3b*). By wall, we mean that the spread of EC regions along a (curvilinear) plane is broader than that along its normal direction. We can discern this feature clearly from the A-only snapshots shown for $\Lambda = 0.6188$ in *Figure 3b*. Predominantly, the A's form a wall-like spherical shell, and as per the given composition of the system, the rest of the A's too arrange themselves in the wall-like manner. This kind of organization is in sharp contrast with surface-minimizing globule-shaped organization of beads in response to affinity-driven phase separations. This is a robust feature we note in all of our simulations for different $\phi_A$ (0.35, 0.40, 0.50, 0.60, and 0.70) and $\epsilon_{HC}$ (0, 2, 4, and 6) in the presence of enzymatic activity. *Figure 3* exemplifies the results for $\phi_A = 0.40 - 0.60$ and $\epsilon_{HC} = 4$, and also demonstrates a phase separation without enzymatic activity.

Next, we examine the orientation of the chromatin segments in the wall-like organization of the EC regions by measuring a local nematic order of the AA bonds (see Methods). Within coarse-grained localities, the AA bonds show negative nematic order parameter in the presence of the enzymatic activity, while no significant order is observed for $\Lambda = 0$ (*Figure 3—figure supplement 1*). The negative nematic order of AA bonds implies that those bonds are approximately parallel to the plane along the wall (*de Gennes and Prost, 1993*). We calculate the mean local nematic order parameter of the AA bonds, $S_{AA}$, averaged across the cavity except near the surface (see Methods), and plot it in *Figure 3c*. The consistent trend of negative $S_{AA}$ for $\Lambda \neq 0$ portrays the association of the characteristic wall-like organization and the local order of the AA bonds therein (see *Figure 3—figure supplement 2* for phase diagrams on $\Lambda - \epsilon_{HC}$ plane for several $\phi_A$'s).

We also calculate the mean local nematic order parameter $S_{BB}$ of the BB bonds in the system and note that $S_{BB} > 0$ for $\Lambda > 0$ (see *Figure 3—figure supplement 2*). However, the BB bonds do not show nematic ordering in the absence of enzymatic activity, even for $\epsilon_{HC} > 2$ where the system phase separates due to HC affinity. Combining our observations of negative nematicity of AA bonds and positive nematicity of BB bonds in the phase separated systems for $\Lambda > 0$ in contrast to the HC affinity-induced phase separation, we conclude that the enzymatic activity breaks the local isotropy of the bonds.

The emergence of the above-mentioned characteristic features due to Topo-II activity, especially the non-globule wall-like morphology of EC regions with associated negative nematic order of the AA bonds therein, are non-trivial findings of our research. To understand specifically what aspect of the proposed model resulted in such characteristic features, here we discuss the results obtained for a few variant models of MdDAM. First, it is important to note that the wall-like morphology has not been observed in simulation settings with non-transient reduction of steric repulsion potentials among one type of beads, as in our self-avoiding-phantom-polymer model (*Figure 2a*) and in *Fujishiro and Sasai, 2022*. In addition, we note that the A'A' bonds do not exhibit negative nematic order in the self-avoiding-phantom-polymer model (*Figure 3—figure supplement 3e*). We also considered a non-transient effective attraction model that may mimic the transient attraction state between A beads bound to an enzyme in MdDAM and found no phase separation at all for the case comparable to $\Lambda = 0.0309$ (*Figure 3—figure supplement 4*). Taken together, it seems that the characteristic features observed in MdDAM are rooted in the transient nature of the enzymatic activity.

To further investigate which transient interaction in our Topo-II model is essential for the emerged characteristic features, next we study two other transient variants of MdDAM. MdDAM is a three-state transient model where the interaction $h_{vex}$ among a chosen pair of proximal A beads switches from repulsion to attraction to no interaction to repulsion state (RANR, *Figure 3—figure supplement 3*). In contrast to that, first we consider a two-state transient model RAR where $h_{vex}$ switches from repulsion to attraction to repulsion state. For this alternative model, we note phase separation configurations similar to that in MdDAM with the characteristic features mentioned above. We study another

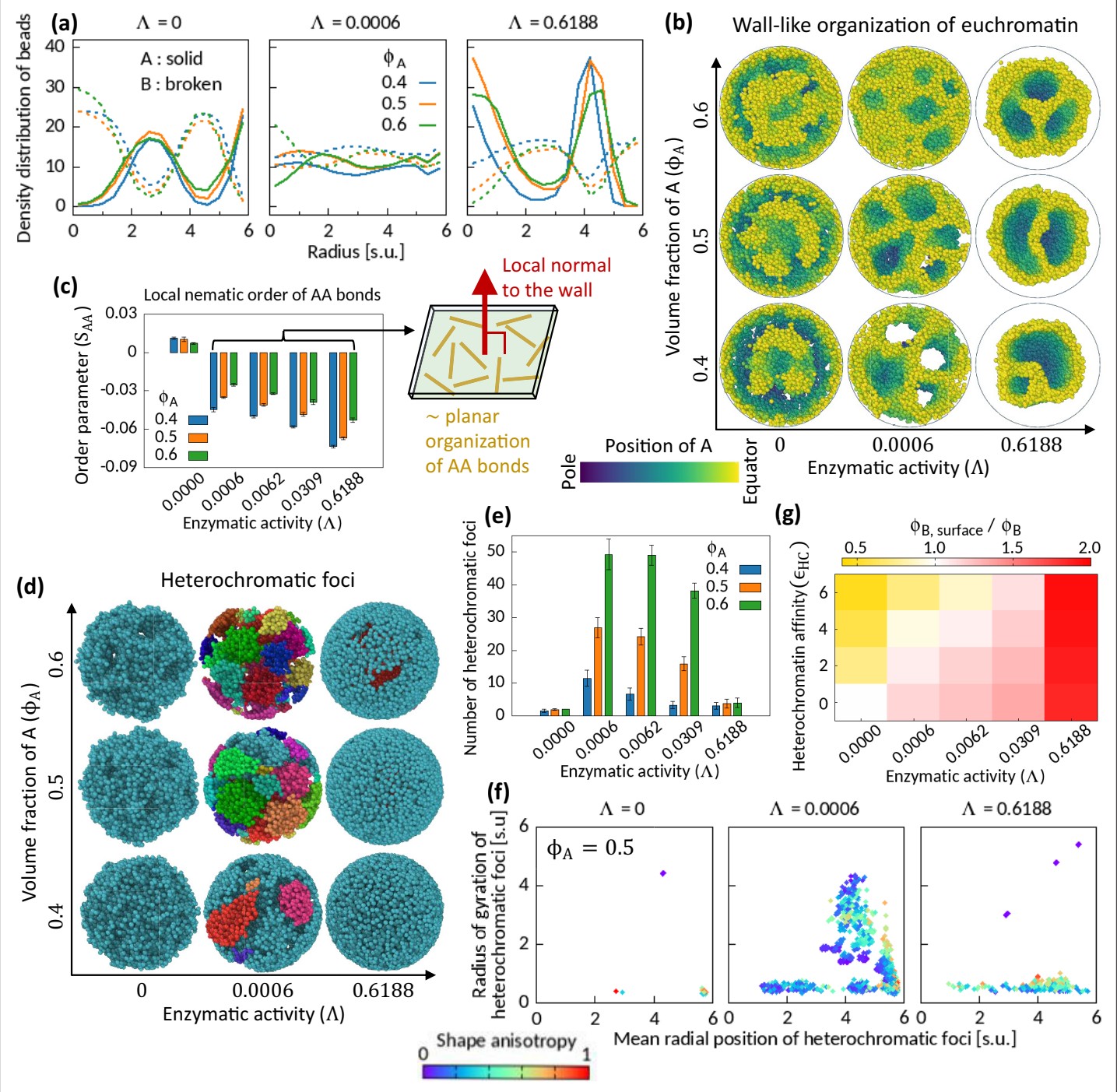

**Figure 3.** Characteristics of Topo-II-induced microphase separation configurations. (**a**) Density distribution of A and B beads in radial direction, plotted for fixed $\epsilon_{HC} = 4$. (**b**) Sample snapshots (hemisphere cuts) showing wall-like organization of A's for $\Lambda > 0$. (**c**) Local nematic order parameter of AA bonds. Schematic shows approximate organization of AA bonds in the wall. (**d–f**) Heterochromatic foci features. Sample snapshots (**d**) and number (**e**) of heterochromatic foci are shown. In (d), B beads (heterochromatin) are shown, where different color of the beads indicates distinct focus. Time-averaged data are shown in (**e**) and the error bars indicate standard deviations over four realizations. (**f**) Position and size of individual focus are respectively represented by the mean radial coordinates of the member-B's and the radius of gyration of the focus. Shape anisotropy ranges from zero to unity for spherical and line-shaped foci, respectively. (**g**) Volume fraction of B's at the surface over the global volume fraction of B is shown in $\Lambda - \epsilon_{HC}$ space for $\phi_A = 0.5$. Time-averaged data are shown.

The online version of this article includes the following figure supplement(s) for figure 3:

**Figure supplement 1.** AA bonds along the polymer are organized in planes of the walls formed by Topo-II.

*Figure 3 continued on next page*

*Figure 3 continued*

**Figure supplement 2.** Topo-II breaks local isotropy of the AA and BB bonds.

**Figure supplement 3.** Comparison of monodisperse differential active model (MdDAM) with other transient models.

**Figure supplement 4.** A non-transient effective attraction model comparable to monodisperse differential active model (MdDAM) does not show phase separation.

**Figure supplement 5.** Effect of the geometry of the cavity on phase separation configurations.

**Figure supplement 6.** Features of heterochromatic foci shown for $\epsilon_{HC} = 4$.

two-state transient model RNR where $h_{vex}$ switches from repulsion to no interaction to repulsion state. From their visual appearance, the phase separation configurations in the RNR model seem similar to that observed in MdDAM; however, their $P(\Delta\phi)$ are very different, and the AA bonds do not show nematic ordering in the RNR model. Altogether, we may conclude that (i) the transient feature of enzymatic activity is essential to wall-like spatial arrangement of A beads and (ii) a transient attraction state is necessary for the negative nematicity of the AA bonds in the phase separation configurations. Note that, thus, the transient pair-catching nature is likely essential for generation of wall-like features, implying that the wall-like features can result from not only by the Topoisomerase-II activity but also by other types of enzymes with transient pair-catching nature. We hope that further research will be carried out using different types of enzymes.

We have checked that the spherical appearance of the walls near the surface of the cavity is due to our choice of the cavity geometry. We have simulated MdDAM for the case of a cubic geometry with closed boundary in one direction (analogous to the radial confinement in our main model) and periodic boundaries in other two directions. For $\Lambda > 0$, there we see wall-like appearance of the EC domains with the associated negative nematic order of AA bonds therein. Near the closed boundaries, the walls are parallel to that plane (see *Figure 3—figure supplement 5*).

In our simulations, we also assumed that the pair of the beads caught by Topo-II are not the immediate neighbors along the polymer. We performed the simulation without this assumption and observed that the obtained phase separation configurations and the nematic order parameters of the AA and the BB bonds (for $\phi_A = 0.5$, $\epsilon_{HC} = 0.0$ and $\Lambda = 0.6188$, $S_{AA} = -0.0514$ and $S_{BB} = 0.2957$) compare well with our original model ($S_{AA} = -0.0726$ and $S_{BB} = 0.3060$).

## Effect of Topo-II on HC foci

For the cell to function properly, the number, size, and spatial position of HC foci have to be critically regulated (*Fodor et al., 2010*; *Wang et al., 2018*; *Elgin and Reuter, 2013*). So, we segmented HC foci (see Methods) from the simulated snapshots and investigate their features to understand the role of Topo-II's activity on them. We show sample snapshots of segmented foci on *Figure 3d*. Most of the B's remain connected under the action of HC affinity and in the absence of enzymatic activity (*Figure 3d*). Consequently, we count a small number of foci (*Figure 3e*)—a relatively big sponge-like focus spread across the cavity, and a few other small foci scattered elsewhere (*Figure 3f*). For strong enzymatic activity, we count a small number of foci with mainly two dominating modes—one, localized near the surface of the cavity having various sizes depending on $\phi_A$, and second, a focus localized inside the cavity having a notable morphology (*Figure 3d–f* and *Figure 3—figure supplement 6*). For moderate enzymatic activity, we see many foci of various shapes and sizes. The scatter plot in *Figure 3f* is color coded by the shape anisotropy (see Methods) of the individual foci, and it suggests the appearance of foci with various shapes.

## Topo-II in determining surface profile of chromosome territory

In conventional nucleus, HC regions accumulate near the nuclear membrane, whereas transcriptionally active genes mostly localize at the intermingling regions of two chromosome territories (*Falk et al., 2019*; *Uhler and Shivashankar, 2017*; *Shivashankar, 2019*). Thus, there exists an orchestration of mechanisms that determines whether EC and/or HC regions will localize at the surface of a chromosome territory. While a combination of strong HC affinity (*Larson et al., 2017*; *Strom et al., 2017*), and interaction between lamina and HC contents (*Briand and Collas, 2020*; *Guelen et al., 2008*) can explain this phenomenon, one cannot exclude the possibility of other mechanisms playing a significant role in this regard. We note that the enzymatic activity of Topo-II competes with HC affinity in

determining surface localization profile. HC affinity pulls B's inward the cavity to minimize the inter-facial energy cost. On the other hand, enzymatic activity drives A's inside the cavity. To illustrate this competition, we calculate the ratio of the mean volume fraction of B's at the surface of the cavity ($\phi_{B,surface}$, see Methods) to the global volume fraction of B's, $\phi_B = 1 - \phi_A$, and construct a heatmap (**Figure 3g**). The heatmap manifests this competition and hints at the existence of an isoline on the $\Lambda - \epsilon_{HC}$ plane where $\phi_{B,surface} = \phi_B$. In **Figure 3g**, we show the heatmap for the symmetric composition; however, we find similar heatmap for other $\phi_A's$.

## Bidisperse chromatin model

A recent super-resolution microscopy study showed that at the epigenetic level, histone marks characterizing EC and HC regions remain at different structural states (**Xu et al., 2018**). Active histone marks like H3K9ac form distinct and small clusters as compared to condensed large aggregates of the repressive marks such as H3K27me3. An implication of this experimental observation from the polymer physics model perspective is that the beads representing EC and HC regions have different sizes. Thus, we came up with the idea of bidisperse chromatin. Bidispersity is known to affect the phase separation pattern of colloidal systems (**Hueckel et al., 2021**; **Asakura and Oosawa, 1954**; **Vrij, 1976**). Here, we study the effect of enzymatic activity on bidisperse chromatin. We modify our copolymer model in such a way that the A and B bead sizes are different from each other and respectively equal to the mean sizes of H3K9ac- and H3K27me3-clusters (**Figure 4a–b**). Hereafter, we refer this modified setting as *bidisperse differential active model* (BdDAM). To simulate this system for a biologically relevant composition, we extracted the radial distribution function data from **Xu et al., 2018**, and obtained the corresponding volume fraction as $\phi_A = 0.3544$ (**Figure 4a**; also see Methods).

We first simulate BdDAM without any HC affinity and enzymatic activity and obtain the order parameter distribution $P(\Delta\phi)$ (**Figure 4c**—bottom; red curve with open square symbol). We note that bidispersity alone can drive MPS in the system, which is also evident from the sample snapshot shown in **Figure 4d** for the case of $\epsilon_{HC} = 0$ and $\Lambda = 0$. This phase separation and the localization of the bigger beads (HC) at the surface of the cavity are driven by the depletion forces (**Asakura and Oosawa, 1954**; **Vrij, 1976**). Next, we focus on the effect of the enzymatic activity on this bidisperse setting keeping $\epsilon_{HC} = 0$. In the presence of enzymatic activity, we see MPS phenomenology with similar characteristic features observed for the MdDAM case, viz., asymmetric profile of $P(\Delta\phi)$ (**Figure 3c**) and wall-like organization of EC regions (**Figure 3d**) with the associated negative nematic order of the AA bonds (**Figure 3e**). To compare the obtained MPS configurations for BdDAM with the corresponding MdDAM case, we calculate density-density cross-correlation between the two models (shown for A beads in **Figure 4f**; also see Methods). The results imply a strong correlation between two model configurations under the action of enzymatic activity.

We also investigate the case in the presence of all—enzymatic activity, HC affinity, and bidispersity. Even with the HC affinity, the tendencies similar to those mentioned right above are retained (**Figure 4c–f**; $\epsilon_{HC} = 6$). Therefore, we conclude that enzymatic activity affects the phase separation phenomenology in a similar way for both the monodisperse and the bidisperse setting. However, we also note that there is still a visible difference in the MPS morphology due to bidispersity (**Figure 4d**).

## Discussion

In summary, we have investigated the role of Topo-II in interphase chromatin organization using a random copolymer model with coarse-grained blocks representing EC and HC regions, where Topo-II drives the system out of equilibrium. We noted that Topo-II has an intrinsic ability to microphase separate the chromatin. To understand the underlying mechanism of this phase separation, we studied a simplified equilibrium polymer model as well as a simplified mean-field framework. These studies suggest that transient phantomness of a subsection of polymer due to Topo-II activity can drive this phase separation. However, in spite of being the essential mechanism for phase separation, it does not explain the characteristic wall-like organization of the EC regions that emerge due to precise mechanistic scheme of Topo-II activity. Further, exploiting our polymer model, we show that bidispersity of chromatin due to different sizes of epigenetic marks affects its MPS morphology, however, the characteristic features of Topo-II-induced MPS survive there.

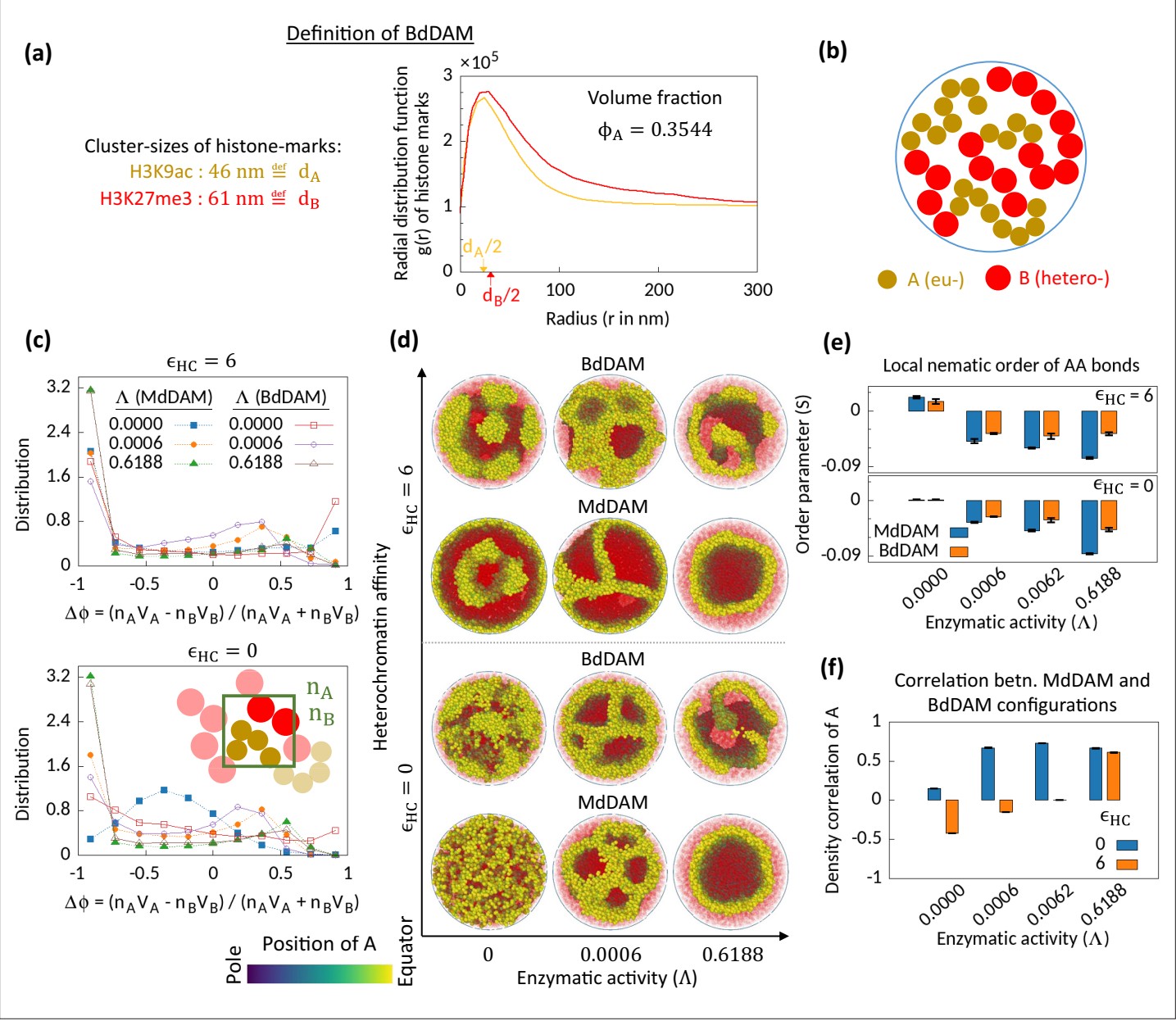

**Figure 4.** Microphase separation in bidisperse model, motivated from super-resolution microscopy data. (**a**) Extracted data for mean cluster sizes and radial distribution functions of histone marks characteristic to eu- and heterochromatic regions. The data were extracted from **Xu et al., 2018**. Following the experimental data, we set volume fraction $\phi_A = 0.3544$. (**b**) Schematic of the bidisperse random multiblock copolymer model. (**c–f**) Comparison of the bidisperse differential active model (BdDAM) with the monodisperse differential active model (MdDAM). Time-averaged $\Delta\phi$-distributions are shown in (**c**), and the error bars over realizations are not shown as those are smaller than the symbol sizes. (**d**) Sample snapshots (hemisphere cuts) are shown, where the B's are represented in semi-transparent red. The bidisperse system shows phase separation even for $\epsilon_{HC} = 0$ and $\Lambda = 0$. (**e**) Local nematic order parameters, averaged over realizations, are shown and the error bars indicate the corresponding standard deviations. (**f**) Cross-correlation of local density of A's between MdDAM and BdDAM configurations are shown (see Methods for definition). The data shown are averaged over four realizations, and the error bars indicate the corresponding standard deviations.

Recently, chromatin organization has been investigated extensively in the context of phase separation, and several different mechanisms and models have been proposed. In **Fujishiro and Sasai, 2022**, repulsion-driven phase separation was proposed using polymer physics model simulations. It was assumed that the chromatin configuration in EC regions is looser and more flexible than in HC regions which allowed for the overlap between the monomers representing EC regions and led to the phase separation of the monomers representing EC and HC regions. Our equilibrium

self-avoiding-phantom-polymer model may be regarded as a limit of their repulsive-interaction-only polymer model.

In *Heo et al., 2021*, the authors developed a field-based model simulating HC formation which incorporates the kinetics of methylation and acetylation in order to clarify the impact of the changes in histone methylation status on chromatin condensation. They found that the methylation/acetylation reactions lead to interconversion of the EC and HC phases, and it provides more HC-EC interfaces. For the simulations performed in the present paper, we have used histone marks to designate EC and HC regions, and hence they are not changing in time. However, kinetics of histone mark alteration can induce phase separation. A possible future direction may be to integrate such reaction kinetics of histone modification into our polymer-based model and study how it can interplay with the enzymatic activity-induced phase separation in determining the HC-EC interface property.

Our simulations as well as the aforementioned works focused on the EC and HC phase separation in the nucleus. On the contrary, *Amiad-Pavlov et al., 2021*; *Bajpai et al., 2021*, investigated the chromatin-aqueous phase separation. They found near-surface organization of the entire chromatin content. In our studies, we assumed high packing-fraction situation which does not allow for such chromatin-aqueous phase separation. However, we expect that our model will reproduce near-surface organization of both HC and EC if we extend it by considering a larger cavity, incorporating chromatin-lamin interactions, and tuning parameters like a polymer in bad solvent case (i.e., setting a strong inter-bead attraction).

We saw that Topo-II-induced MPS causes the wall-like appearance of EC domains. This is unlikely to be explained by another mechanism of phase separation proposed for chromatin organization in literature (*Agrawal et al., 2020*; *Ganai et al., 2014*), which relies on inhomogeneous effective temperature. Inhomogeneous temperature models are essentially out-of-equilibrium whereas, in the mechanism which we propose, the MPS itself happens even in equilibrium as suggested by the self-avoiding-phantom-polymer model study. Therefore, our study highlights the importance of mechanistic models to understand the influence of out-of-equilibrium biophysical processes in chromatin organization.

In the current model, we have assumed that the enzymes act homogeneously across the cavity. However, there may exist a spatial distribution profile of the enzymes' action sites, as is the case for RNA polymerases localized within transcription factories (*Iborra et al., 1996*). Moreover, our study has focused on Topo-II enzyme only, whereas it is more likely that the precise mechanistic scheme of other type of enzymes present in the nucleus would significantly affect the chromatin organization. Therefore, a more comprehensive model overcoming the above-mentioned limitations would be beneficial to the field. Nevertheless, even our simple model study reveals that the mechanistic scheme of enzymatic activity plays a critical role in determining spatial features of eu- and heterochromatin architectures.

In general, there are a number of situations where heterochromatin architecture changes depending on the state or condition of a cell. For example, aging correlates with the heterochromatin architecture (*Tsurumi and Li, 2012*; *Lee et al., 2020*). Also, in aging, activity of various enzymes is known to undergo profound changes with cell state modifications (*Adelman and Britton, 1975*; *Becker and Rudolph, 2021*). A part of aging-associated alteration of heterochromatin architecture might be attributed to the variation of enzymatic activity. Furthermore, alteration of heterochromatin architecture is observed for other cell state modifications like cell differentiation and under external forcing, although less is known about variation of enzymatic activity in those cases (*Meshorer et al., 2006*; *Talwar et al., 2013*; *Damodaran et al., 2018*). Our finding suggests that further experiments focusing on the correlations between enzymatic activity and chromatin organization would provide hints to find out the mechanisms of such alteration of heterochromatin architecture and hence the cell state-specific genome regulation.

## Methods
### Simulation setting for the monodisperse model

We simulate a 50.807 Mbp long single chromatin packed within a spherical cavity of diameter $d_{ct} = 1.4112$ μm. The size of the chromatin territory (i.e., the spherical cavity) was chosen to comply with a typical density of human diploid genome where 6.2 Gbp DNA is packed within a nucleus of

diameter 7 µm. This single chromatin is mimicked by a bead-and-spring model comprising $N = 20992$ equal-sized beads. Therefore, each bead represents 2.4203 kbp chromatin. We assume nucleosomes as spheres of diameter $d_{nucleosome} = 22$ nm (histone octamer core plus the linker DNA) containing 200 bp of DNA. We further assume close compaction of nucleosomes within the beads A and B, such that $d_A = d_B = d_{nucleosome} \times (number\ of\ nucleosomes\ per\ bead)^{1/3}$. This determines the diameter of the beads as $d_A = d_B = 50.5086$ nm for the monodisperse model.

In our active polymer model, we keep $\lambda_{an} = 16.7\,\tau^{-1}$ and $\lambda_{nr} = 500\,\tau^{-1}$ fixed, $\tau$ being the unit time in our simulation. Note that for the above choice of the rates $\lambda_{an}$ and $\lambda_{nr}$, a pair of beads bound to a Topo-II on average spends $0.0599\,\tau$ in the attraction state and $0.0020\,\tau$ in the no interaction state. We have checked that by the time $(1/\lambda_{an} + 1/\lambda_{nr})$, the mean square displacement of a typical bead is of the order of the bead size. The rate $\lambda_{ra}$ is treated as a simulation parameter.

## Simulation setting for the bidisperse model

We set sizes of the beads, A and B, same as the mean sizes of the histone mark clusters, H3K9ac and H3K27me, respectively; therefore, $d_A = 46$ nm and $d_B = 61$ nm (**Xu et al., 2018**).

To obtain a biologically relevant composition parameter (i.e., $\phi_A$ in our model), we extracted the radial distribution function (RDF, $g(r)$) data for the histone marks H3K9ac and H3K27me3 from **Xu et al., 2018**. Those RDFs were calculated by averaging over several two-dimensional segments of the captured microscopy images. We calculated $m_A = \int_{segment} 2\pi r g_A(r)dr / \int_0^{d_A/2} 2\pi r g_A(r)dr$ (similarly $m_B$) and obtained the volume fraction of the A beads as $\phi_A = m_A V_A/(m_A V_A + m_B V_B) = 0.3544$.

We simulate the bidisperse model with the above-mentioned bead sizes and volume fraction using the length of the polymer $N = 17,664$ that keeps the total DNA content (in bp) same as the monodisperse model.

## Simulation units

We set $d_{ct} = 12\,\ell$ that gives us the simulation unit (s.u.) of length as $\ell = 117.6$ nm. Our model is simulated at a physiological temperature $T = 310$ K, and we consider energy unit as $e = 1\,k_BT = 4.28$ pN·nm. The frictional drag on monomers is approximated by Stokes' law, and the corresponding viscosity (of nucleoplasm) is assumed to be 1.5 cP (**Liang et al., 2009**). Considering the nucleoplasmic viscosity as unity in simulations, we get the simulation time unit $\tau = 0.57$ ms.

## Brownian dynamics

The position $\boldsymbol{x}_i$ of the $i$th bead is updated by integrating

$$\partial_t \boldsymbol{x}_i = -\frac{1}{6\pi\eta(d_i/2)}\partial_{\boldsymbol{x}} H_i + \sqrt{\frac{2k_BT}{6\pi\eta(d_i/2)}}\zeta, \tag{1}$$

where $\zeta$ represents a univariate white Gaussian noise with zero mean, and $\eta$ represents the nucleoplasmic viscosity. $H_i$ represents the total potential energy that the bead realizes in the system, which is given by

$$H_i = H_{spring} + H_{vex} + H_{HC} + H_{confinement}. \tag{2}$$

We explain the terms on the right-hand side of **Equation 2** sequentially in the following paragraphs. To numerically integrate **Equation 1** with **Equation 2**, time is discretized into steps as usual, and the positions $\boldsymbol{x}_i$ of all the beads are updated sequentially over steps.

The potential energy associated with the spring connecting two consecutive beads along the polymer is given by $h_{spring} = -\frac{1}{2}kr_0^2\ln\left[1 - (r_{ij}/r_0)^2\right]$, where $r_{ij}$ is the distance between the $i$th and $j$th beads. We consider $H_{spring} = \sum_{j(\in n.i)} h_{spring}$, where the summation $\sum_{j(\in n.i)}$ runs over the beads next to $i$ (i.e., $j$ takes $i-1$ and $i+1$ if the $i$th bead is not located at one of the polymer ends, whereas it takes only either of them if $i$th bead is located at an end). Here, $k$ is the spring constant of a finitely extensible nonlinear elastic spring whose stretch $r_{ij} \leq r_0$.

When a pair of beads separated by $r_{ij} \leq \ell$ is *not bound* to a Topo-II enzyme, each bead realizes steric repulsion due to one another. We consider that interaction potential between those two beads as

$$h_{vex} = \epsilon_{vex}\exp\left(-\alpha_{vex}r_{ij}^2\right), \text{ when bead-pair } (ij) \text{ not bound to Topo-II}, \tag{3}$$

and the total steric potential realized by the $i$th bead is $H_{vex} = \sum_{j \in r_{ij} \le \ell} h_{vex}$. Here, $\epsilon_{vex}$ is the amplitude of the Gaussian interaction potential chosen, and $\alpha_{vex}$ determines its variance. We set sizes of the beads in both the monodisperse and the bidisperse model by setting a criterion that the minimum of $h_{spring} + h_{vex}$ appears at $r_m = \sum_{i \in \text{connected beads}} d_i/2$, $i \equiv$ A and/or B. This sets a constraint over the choice of the parameters in $h_{spring}$ and $h_{vex}$ as $kr_m^2 / \left(1 - r_m^2/r_0^2\right) = 2\epsilon_{vex}\alpha_{vex}r_m^2 \exp\left(-\alpha_{vex}r_m^2\right) = G$, say. Note that $G$ is dimensionless, and we keep it fixed for all the choices of the model parameters in the monodisperse and the bidisperse model.

As per our active polymer model described in the main text, if a pair of beads ($ij$) separated by $r_{ij} \le \ell$ is *bound* to a Topo-II enzyme, the beads either realize attraction for each other, or there is no steric interaction among those beads (*Figure 1b*). We implement this model scheme by tuning $h_{vex}$ as following:

$$
\begin{aligned}
h_{vex} &= -\epsilon_{vex}\exp\left(-\alpha_{vex}r_{ij}^2\right), \text{ when bead-pair ($ij$) bound to Topo-II and in the attraction state,}\\
&= 0, \text{ when bead-pair ($ij$) bound to Topo-II and in the no-interaction state.}
\end{aligned}
\tag{4}
$$

The interaction potential $h_{vex}$ between a pair of beads ($ij$) separated by $r_{ij} \le \ell$ ($i$ and $j \equiv$ A, as per our assumption of the current model) starts the following series of Poissonian transitions: state $(h_{vex} > 0) \xrightarrow{\lambda_{ra}}$ state $(h_{vex} < 0) \xrightarrow{\lambda_{an}}$ state $(h_{vex} = 0) \xrightarrow{\lambda_{nr}}$ state $(h_{vex} > 0)$, where each transition step can take place in between any consecutive two steps of $x_i$-dynamics (following *Equation 1* and *Equation 2* discretized in time) and is implemented stochastically according to the Poisson process with the rate $\lambda_{XX}$ ($XX \equiv$ ra, an, nr). When the bead-pair ($ij$) get separated by $r_{ij} \ge \ell$ during this process, those pair of beads temporarily stop proceeding with the stochastic steps of the transitions described above. Note that in our current model, the pairs AB and BB always assume the expression of $h_{vex}$ given in *Equation 3* for a separation $r_{ij} \le \ell$.

If the $i^{th}$ bead is classified as B, it realizes an affinity potential due to other proximal B beads $H_{HC} = \sum_{j \in B} h_{HC}$, where $h_{HC} = -\epsilon_{HC}r_{ij}^2 \exp\left[-\alpha_{HC}\{d_B - 1/(\alpha_{HC}d_B) - r_{ij}\}^2\right]$ when they are separated by $r_{ij} \le \ell$; $h_{HC} = 0$ otherwise. Here, $\epsilon_{HC}$ parameterizes the affinity, which has dimension same as spring constant. $h_{HC}$ has its minimum value at $r_{ij} = d_B$, and $\alpha_{HC}$ determines width of the potential well about its minimum. We choose this functional form of $h_{HC}$ to ensure that its minimum point coincides with the minimum of $h_{spring} + h_{vex}$, and thus, it does not interfere with our scheme to fix bead sizes.

To confine the polymer inside the cavity, we use a potential $H_{confinement}$ that acts upon a bead at a separation $r$ from the wall of the cavity. We consider this potential energy as that between a wall and a star polymer with functionality $s = 2$ (therefore, a linear polymer) (*Jusufi et al., 2001*; *Camargo and Likos, 2009*):

$$
\begin{aligned}
H_{confinement} &= ps^{3/2}\left[-\ln\left(\frac{r}{R_s}\right) - \left(\frac{r^2}{R_s^2} - 1\right)\left(\frac{1}{1 + 2\kappa^2 R_s^2} - \frac{1}{2}\right) + \gamma\right], \text{ for } r \le R_s,\\
&= ps^{3/2}\gamma\,\text{erfc}\left(\kappa r\right)/\text{erfc}\left(\kappa R_s\right), \text{ for } r > R_s.
\end{aligned}
\tag{5}
$$

**Table 1.** Choice of model parameters.

| Potential | Parameters |
|---|---|
| $H_{spring}$ | $k = 22\, e\ell^{-2}$; $r_0 = \sum_{i \in \text{connected beads}} d_i$. |
| $H_{vex}$ | Monodisperse model: $\epsilon_{vex} = 8\,e$; $\alpha_{vex} = 7.9585\,\ell^{-2}$. |
| | Bidisperse model for AA pairs: $\epsilon_{vex} = 6.6539\,e$; $\alpha_{vex} = 9.5686\,\ell^{-2}$. |
| | Bidisperse model for BB pairs: $\epsilon_{vex} = 11.507\,e$; $\alpha_{vex} = 5.5330\,\ell^{-2}$. |
| | Bidisperse model for AB pairs: $\epsilon_{vex} = 8.9153\,e$; $\alpha_{vex} = 7.1414\,\ell^{-2}$. |
| $H_{HC}$ | $\alpha_{HC} = 100\,\ell^{-2}$. |
| $H_{confinement}$ | $p = 4\,e$; $s = 2$; $R_s = 0.65R_g$; $\kappa = 1/R_g$; $R_g = d_i/2$ where $i \equiv$ A, B |

Here, we have modeled individual bead as a star polymer with radius of gyration $R_g = d_i/2$, $i \equiv$ A, B. In *Equation 5* Brownian dynamics, $p$ is a free parameter, $R_s = 0.65 R_g$ stands for radius of corona of the polymer, the parameter $\kappa$ should be of the order of $R_g^{-1}$, $\gamma = \frac{\sqrt{\pi}\, \mathrm{erfc}(\kappa R_s) \exp(\kappa^2 R_s^2)}{[\kappa R_s (1+2\kappa^2 R_s^2)]}$, and erfc stands for complementary error function.

We summarize our choice of model parameters in *Table 1*.

*Equation 1* is integrated over time in Euler method, where we use a discretized time step $10^{-4}\tau$. The simulations are done using lab-developed codes where we use CUDA to exploit GPU acceleration and OpenMP for CPU parallelization. We start the simulations from equilibrated ideal chain configurations confined within the spherical cavity. The system is annealed for a time span (typically, $2010\tau$) by which the mean square displacement of a bead is more than the radius of the cavity (*Nuebler et al., 2018*). Next, simulations are executed for the same duration as the annealing time, and numerous snapshots are stored. The results presented in the paper are obtained by analyzing such snapshots from at least four different realizations for each set of parameters.

## Quantification of phase separation

The whole cavity is divided into cubic grids of linear size $\ell$, and a parameter $v_i = $ (*volume of $i^{th}$ grid accessible to the beads*)$/\ell^3$ is calculated for each grid. The distribution of the order parameter $\Delta\phi$ is defined as

$$P(\Delta\phi) = \left\langle \left[\sum_{i \in grids} v_i \delta(\Delta\phi - \Delta\phi_i)\right] / \left[\sum_i v_i\right] \right\rangle_{snapshots},$$ (6)

where $\delta$ indicates Dirac delta function. The Binder cumulant and the moment coefficient of skewness of $\Delta\phi$ are defined as $1 - \langle(\Delta\phi)^4\rangle_P / 3\langle(\Delta\phi)^2\rangle_P^2$ and $E\left[(\Delta\phi - \langle\Delta\phi\rangle_P)^3\right] / \left\{E\left[(\Delta\phi - \langle\Delta\phi\rangle_P)^2\right]\right\}^{3/2}$, respectively. Here, $E[\cdot]$ signifies the expectation operator.

## Mean-field framework

We consider a lattice space of size $M$ containing $M_{A'}$ number of A' and $M_B$ number of B beads, such that $M = M_{A'} + M_B$. Two A' beads can overlap to form a doublet (D) leaving an empty (E) site. We define the doublet fraction $\alpha = $ (number of D's)$/M_{A'}$, and therefore $0 \leq \alpha \leq 1/2$. Defining the volume fraction of B in this lattice model as $\Phi_B = M_B/M$, we can write the volume fractions of A', D, and E as $(1 - \Phi_B)(1 - 2\alpha)$, $(1 - \Phi_B)\alpha$, and $(1 - \Phi_B)\alpha$, respectively. Given this setting, we can write the free energy density of the system as

$$
\begin{aligned}
f(\Phi_B, \alpha, \{\epsilon\}) &= \Phi_B \ln \Phi_B + (1 - \Phi_B) \ln(1 - \Phi_B) \\
&+ (1 - \Phi_B)\left[2\alpha \ln \alpha + (1 - 2\alpha) \ln(1 - 2\alpha)\right] + f_{int}(\Phi_B, \alpha, \{\epsilon\}),
\end{aligned}
$$ (7)

where $\{\epsilon\}$ represents the pair interaction strengths among A', D, E, and B, and $f_{int}$ stands for the total interaction energy. We set $\epsilon_{BB} = \epsilon$, $\epsilon_{A'B} = \epsilon$, $\epsilon_{DB} = 2\epsilon$, $\epsilon_{A'D} = \epsilon$, $\epsilon_{DD} = 2\epsilon$, and rest of the pair interactions are set to zero. This choice of the interaction parameters gives us

$$f_{int}(\Phi_B, \alpha, \{\epsilon\}) = \epsilon\left[c(\alpha)\Phi_B^2 - 2c(\alpha)\Phi_B + \left(c(\alpha) + \tfrac{1}{2}\right)\right],$$ (8)

where $c = -\alpha^2 + \alpha - 1/2$. Minimizing $f$ with respect to $\alpha$, we obtain the critical doublet fraction $\alpha^*(\Phi_B, \epsilon)$, and thereby we obtain the reduced free energy density $f^*(\Phi_B, \alpha^*, \epsilon)$. Note that we have considered only repulsive interactions among the lattice pairs, and therefore, we call $\epsilon$ a pair repulsion parameter.

## Nematic order parameter

The whole spherical cavity is gridded into cubic localities with lateral dimension $\ell (= d_{ct}/12$ and $> 2d_{A,B})$. We define a specific type of bond (i.e., AA or BB) as $\boldsymbol{u} = \boldsymbol{x}_{i+1} - \boldsymbol{x}_i$, $i \in [1, N]$. Given multiple ($>4000$) snapshots at equally separated time points for a realization, we count the total number of specific type of bonds $q_j$ in the locality $j$. We construct the local nematic tensor $\boldsymbol{Q}_j = (3\boldsymbol{u}_j \otimes \boldsymbol{u}_j - \boldsymbol{I})/2$ and diagonalize it. The eigen value of $\boldsymbol{Q}_j$ which has the largest absolute value among three is defined as the local nematic order parameter $S_{j,\delta\delta}$, where $\delta\delta$ indicates specific type of bonds. Grid-wise local nematic

order parameters are shown in *Figure 3—figure supplement 1* for two sample cases. Note that the confinement induces a local nematic order of bonds near the surface, but we are interested to see order emerging due to enzymatic activity. So, we calculate the mean local nematic order parameter of $\delta\delta$ bonds as $S_{\delta\delta} = \sum_{j\notin surface} q_j S_{j,\delta\delta} / \sum_{j\notin surface} q_j$. Here, we consider the outermost spherical shell of width $\ell$ as the surface region of the cavity.

## Segmentation of HC foci and analysis

We load the coordinates of B's on OVITO (*Stukowski, 2010*), an open visualization tool, and use its cluster analysis modifier. Two B's separated by less than or equal to the bead size, $d_B$, are considered to be the members of the same cluster. Any cluster comprising at least $(2 \times block\, size) = 8$ B's are considered as HC focus, otherwise neglected as noise.

To quantify the size and the shape of the segmented HC foci, we calculate gyration tensor of individual focus, defined as $G_{mn} = \frac{1}{n_B} \sum_{i=1}^{n_B} r_m^{(i)} r_n^{(i)}$, where $r_m^{(i)}$ is the $m$th Cartesian coordinate of the member $i$ of the $n_B$ B's forming the focus in its center-of-mass frame. Diagonalizing $\boldsymbol{G}$, we obtain three eigen values, $\lambda_m^2$, $m = x, y, z$, along three principal axes of the focus. The radius of gyration of the focus is given by $\sqrt{\sum_{m=x,y,z} \lambda_m^2}$, and the shape anisotropy is defined as $\kappa^2 = \frac{3}{2} \frac{\sum_{m=x,y,z} \lambda_m^4}{\left(\sum_{m=x,y,z} \lambda_m^2\right)^2} - \frac{1}{2}$, which will be zero for a spherical focus and unity when all the member-B's will fall on a straight line.

## Surface localization profile calculation

We consider the outermost spherical shell of width $\ell$ as surface region. We calculate the ratio of the number of B's to the total number of beads in that shell, and average over multiple snapshots and realizations to obtain $\phi_{B,surface}$.

## Density-density cross-correlation

To compare the configurations obtained for MdDAM and BdDAM, we grid the whole cavity into cubic localities of lateral size $\ell$, and calculate the local density of A's. For a given snapshot of a specific model, if $c_j$ be the density at locality $j$, and $\bar{c}$ is the mean density therein, then we calculate the cross-correlation as

$$\left\langle \frac{\sum_j \left[ (c_{j,MdDAM} - \bar{c}_{MdDAM})(c_{j,BdDAM} - \bar{c}_{BdDAM}) \right]}{\sqrt{\sum_j (c_{j,BdDAM} - \bar{c}_{BdDAM})^2 \sum_j (c_{j,BdDAM} - \bar{c}_{BdDAM})^2}} \right\rangle_{pair\, of\, snapshots} .$$

## Acknowledgements

We thank Andrew Wong and Shu Chian Tay from Mechanobiology Institute (MBI) science communication core for editing the manuscript and MBI computational core for supporting us about computer-related research activities. We also appreciate Yuting Lou for valuable discussions. This research was supported by Seed fund of Mechanobiology Institute (to JP, TH) and Singapore Ministry of Education Tier 3 grant, MOET32020-0001 (to GVS, JP, TH) and JSPS KAKENHI No. JP18H05529 and JP21H05759 from MEXT, Japan (to TS).

## Additional information

### Funding

| Funder | Grant reference number | Author |
|--------|------------------------|--------|
| Mechanobiology Institute, Singapore | Seed grand | Jacques Prost<br>Tetsuya Hiraiwa |
| Ministry of Education - Singapore | Tier 3 grant MOET32020-0001 | GV Shivashankar<br>Tetsuya Hiraiwa<br>Jacques Prost |
| Japan Society for the Promotion of Science | KAKENHI JP18H05529 | Takahiro Sakaue |

| Funder | Grant reference number | Author |
|---|---|---|
| Ministry of Education, Culture, Sports, Science and Technology | JP21H05759 | Takahiro Sakaue |

The funders had no role in study design, data collection and interpretation, or the decision to submit the work for publication.

## Author contributions

Rakesh Das, Conceptualization, Data curation, Formal analysis, Investigation, Visualization, Methodology, Writing - original draft, Writing – review and editing; Takahiro Sakaue, Funding acquisition, Methodology, Writing – review and editing; GV Shivashankar, Conceptualization, Funding acquisition, Writing – review and editing; Jacques Prost, Conceptualization, Supervision, Funding acquisition, Methodology, Writing – review and editing; Tetsuya Hiraiwa, Conceptualization, Supervision, Funding acquisition, Methodology, Writing - original draft, Project administration, Writing – review and editing

## Author ORCIDs

Rakesh Das  http://orcid.org/0000-0002-3233-510X
Takahiro Sakaue  http://orcid.org/0000-0002-0863-6682
Tetsuya Hiraiwa  http://orcid.org/0000-0003-3221-345X

## Decision letter and Author response

Decision letter https://doi.org/10.7554/eLife.79901.sa1
Author response https://doi.org/10.7554/eLife.79901.sa2

# Additional files

## Supplementary files

• Source code 1. CPU-based FORTRAN simulation code using OpenMP API. Instructions to use this can be found in the README text accompanying the source code.

• Source code 2. CUDA FORTRAN simulation code using GPU acceleration. Instructions to use this can be found in the README text accompanying the source code.

• Transparent reporting form

## Data availability

All data generated or analysed are included in the manuscript. Two source codes used to simulate all the variants of models presented here are shared as supplementary files - Source code 1: CPU-based FORTRAN simulation code using OpenMP API. Instructions to use this can be found in the README text accompanying the source code. Source code 2: CUDA FORTRAN simulation code using GPU acceleration. Instructions to use this can be found in the README text accompanying the source code.

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
