## [Editor Report]

This manuscript will be of interest to readers in the field of physical biology and molecular biology for understanding genome organization. The idea of this computational study and its outcomes suggest a novel phase-separated structure and will shed new light on the role of enzymatic activity in chromatin organization. Overall, modeling and simulation are properly performed and analyzed, and the data support the key claims of the manuscript.

---

## [Decision Letter]

**Decision letter after peer review:**

Thank you for submitting your article "How enzymatic activity is involved in chromatin organization" for consideration by *eLife*. Your article has been reviewed by 2 peer reviewers, and the evaluation has been overseen by a Reviewing Editor and Naama Barkai as the Senior Editor. The reviewers have opted to remain anonymous.

The reviewers appreciated the significance of your findings. In particular, the suggested role of topoisomerase activity is very original and the wall structure found is very interesting and novel. Please note, however, the comments below and address them.

*Reviewer #1 (Recommendations for the authors):*

The role of topoisomerase II activity in the spatial organization of chromatin is poorly understood. Typically it is suggested that topoisomerase is required to prevent entanglement of DNA during replication. In the present paper the authors propose a novel idea: That topoisomerase activity in euchromatin would trigger micro phase separation of chromatin in hetero- and euchromatin and could thus play a key role for the spatial organization of chromatin. This is a compelling idea that the authors test using a coarse-grained polymer model of chromatin. Interestingly, in the simulations micro phase separation is observed that is different from equilibrium phase separation in that it generates wall like structures of the phase in which topoisomerase activity is simulated. The model used is simple and elegant, the results interesting and stimulating. However key aspects of the model remain quite unclear.

Key to the work is the coarse-grained polymer model including topoisomerase activity. When reading the paper the description of the model seems incomplete in the main text and somewhat clumsy in the methods part. The model should be more precisely defined so that the model choices are clear and work could be repeated.

– A major point of criticism is that key aspects of the model to describe the topisomerase activity are not properly explained and the model therefore remains unclear. It is not clear how Brownian dynamics is combined with the stochastic transitions at rates λ_an_ and λ_nr_ and λ_ra_ in the simulation. Furthermore the rates λ_an_, λ_nr_ and λ_ra_ are unclear. These rates refer to pairs of beads but it is not explained which pairs of beads are are selected and how. Can these rates be related to the concentration of topoisomerase molecules? Should these not be rates per volume rather than rates?

– The Brownian dynamics is explained in the methods section but as a list of bullet points on p. 14 and 15. Here the model definition and some parameter values are mixed together and some points are rather unclear. Parameter values should be summarized in a table and the contributions to the potential H should be written as equations to improve clarity.

– It seems plausible that the rates λ_an_, λ_nr_ and λ_ra_ can describe the effects of topoisomerase activity as they allow chromatin strands pass each other. However transiently switching off steric repulsion of coarse grained beads has other effects that one may not associate with topoisomerase action. For example transiently removing steric repulsion will affect osmotic compressibility which could also contribute to micro phase separation. It is not clear to what degree the genuine effects associated with topoisomerase activity and other effects that are introduced by the model but could be seen as artefacts contribute relatively to the micro phase separation. In that context there is a short sentence about effects from attraction due to topoisomerase activity: (l. 166) "the possibility … was ruled out". It is unclear to me how this can be "ruled out" and to me this sentence sounds too strong and not clear. A more careful discussion would help clarify these points.

– I am somewhat puzzled by the HC affinity potential described in line 389. It vanishes for r=0 and for large r, so I cannot see that it describes an affinity. What is the physical meaning of epsilon_HC_ and why this choice of potential? In contrast the attractive potential due to enzyme activity is clearly an attractive potential. Why the different choices of potential?

– The quantification of nematic bond order to demonstrate the wall-like nature of microphases is very interesting. However only the nematic order of A-A bonds is discussed. It would be good to show that the B-B bonds do not exhibit similar nematic order.

*Reviewer #2 (Recommendations for the authors):*

This computational work provides a new role of enzymatic activity in chromatin organization, especially Topoisomerase-II (Topo-II). The authors newly introduced a catch-and-release mechanism among euchromatin regions mimicking Topo-II activity and performed simulations of the polymer model. They show that the enzymatic activity promotes the microphase separation of the chromatin model. The model configurations seem consistent with the experimentally observed distribution of euchromatin and heterochromatin. Besides, they provide a theoretical framework for understanding the physical origin of the microphase separation using a simplified mean-field model. The mean-field calculation explains an effective attraction among heterochromatin due to the phantom and self-avoiding contributions, promoting a phase separation. The simulated configurations reveal a characteristic structure called wall-like organization of euchromatin components, which the mean-field framework cannot explain. These data suggest a possibility of forming a wall-like microphase separation in the cell nucleus by enzymatic activity.

1) As polymer modeling approaches have revealed a phase-separated organization such as A/B compartments in the cell nucleus, the existence or assumption of the two type interactions on the active/inactive genomic regions should be a critical factor. This work assumes the catch-and-release mechanism among AA pairs and the attractive interaction among BB pairs. Therefore, the microphase separation would be predictable. However, the wall-like organization is not trivial and might become a universal phase-separated structure in a micro-scale. The outer walls in Figures 1d and 2b seem to be spherical and can be an effect of the spherical boundary condition. The authors do not address the possibility.

2) The reason why the authors change the volume fraction of A and fix the heterochromatin affinity as ε=4 in Figure 3 would be needed to clarify motivation in section "Wall-like organization of EC due to Topo-II."

3) Figure 3a shows the conversion of the A/B compartment configuration due to the enzymatic activity. Then, the authors characterize the wall-like organization of euchromatin by the local nematic order in Figure 3c. How about the local nematic order of BB bonds? The difference would strengthen the wall feature of A compartment regions.

---

## [Author Response]

Reviewer #1 (Recommendations for the authors):The role of topoisomerase II activity in the spatial organization of chromatin is poorly understood. Typically it is suggested that topoisomerase is required to prevent entanglement of DNA during replication. In the present paper the authors propose a novel idea: That topoisomerase activity in euchromatin would trigger micro phase separation of chromatin in hetero- and euchromatin and could thus play a key role for the spatial organization of chromatin. This is a compelling idea that the authors test using a coarse-grained polymer model of chromatin. Interestingly, in the simulations micro phase separation is observed that is different from equilibrium phase separation in that it generates wall like structures of the phase in which topoisomerase activity is simulated. The model used is simple and elegant, the results interesting and stimulating. However key aspects of the model remain quite unclear.Key to the work is the coarse-grained polymer model including topoisomerase activity. When reading the paper the description of the model seems incomplete in the main text and somewhat clumsy in the methods part. The model should be more precisely defined so that the model choices are clear and work could be repeated.

In the updated manuscript, we have revised the description of the simulation model in the main text. The ‘Brownian Dynamics’ part of ‘Methods’ has been rewritten in a lucid manner, where we detail all the interactions in the system using mathematical equations. The choices of the model parameters have been tabulated therein. We believe that the model description is now complete in the revised manuscript, and could be repeated by anyone interested in doing so.

– A major point of criticism is that key aspects of the model to describe the topisomerase activity are not properly explained and the model therefore remains unclear. It is not clear how Brownian dynamics is combined with the stochastic transitions at rates λ_an_ and λ_nr_ and λ_ra_ in the simulation.

The Hamiltonian Hi that the bead-i realizes in the system contains the pairwise steric repulsion potential hvex>0, because all the beads have finite sizes. In our model implementation of Topo-II activity, we assume that the following stochastic transitions happen during the catch-and-release mechanism of Topo-II activity: state(hvex>0)state(hvex<0)state(hvex=0)state(hvex>0). Each of these transition steps can take place in between any consecutive two steps of Brownian dynamics discretised in time, and is implemented stochastically according to the Poisson process with the rate λXX. We have revised our manuscript (Methods section) to ensure that this model description is clear to the readers. ­­

Furthermore the rates λ_an_, λ_nr_ and λ_ra_ are unclear. These rates refer to pairs of beads but it is not explained which pairs of beads are are selected and how.

We have assumed in our active polymer model that Topo-II catches a pair of beads within a spatial proximity r≤ unit length in simulation units (ℓ), and Topo-II molecules are sufficiently abundant so that all the rates are considered uniform across the system.

We have also assumed that the pair of the beads caught by Topo-II are not immediate neighbors along the polymer. We have checked that if we do not make that assumption, the obtained phase separation configurations and the nematic order parameters of the AA and the BB bonds (for ϕA=0.5, ϵHC=0 and Λ=0.6188,
SAA=−0.0514 and SBB=0.2957) compare well with our original model (SAA=−0.0726 and SBB=0.3060). We have made this model choice clear in the revised manuscript and added a statement asserting that this model choice does not affect the main conclusion of our article.

Can these rates be related to the concentration of topoisomerase molecules? Should these not be rates per volume rather than rates?

The rates, λra, λan, and λnr determine the average times a proximal pair of beads spend in the repulsion (Hvex>0), the attraction (Hvex<0), and the no interaction (Hvex=0) states, respectively. While the rates λan and λnr could be understood as coarse-grained parameterization of some intrinsic parameters of the enzyme (therefore, we treated these as given constants), the rate λra could be related to the concentration of the Topo-II molecules. However, we feel these λ-parameters are better represented as *rates instead of rates per volume*, as we do not present a correlation of these λ’s with the Topo-II concentration in our manuscript.

– The Brownian dynamics is explained in the methods section but as a list of bullet points on p. 14 and 15. Here the model definition and some parameter values are mixed together and some points are rather unclear. Parameter values should be summarized in a table and the contributions to the potential H should be written as equations to improve clarity.

We have revised the manuscript following this suggestion.

– It seems plausible that the rates λ_an_, λ_nr_ and λ_ra_ can describe the effects of topoisomerase activity as they allow chromatin strands pass each other. However transiently switching off steric repulsion of coarse grained beads has other effects that one may not associate with topoisomerase action. For example transiently removing steric repulsion will affect osmotic compressibility which could also contribute to micro phase separation. It is not clear to what degree the genuine effects associated with topoisomerase activity and other effects that are introduced by the model but could be seen as artefacts contribute relatively to the micro phase separation.

We thank the editor and the reviewers for raising this point. (Since we are not confident if we could correctly catch what the reviewers mean by the phrase “genuine effect”, we would sincerely appreciate being informed if we are mis-understanding the comments and responding incorrectly).

Reflecting on this comment, especially the last sentence, we thoroughly reconsidered the contents of the manuscript. We realized that, to clarify the genuine effect associated with topoisomerase in a fair and systematic way, we needed clearer statements of which aspect of Topoisomerase-II (or a wider class of enzymes including Topoisomerase-II) is the focus of this study, since generally a model represents only a part of features of a real system. In this study, the focus is on the transient nature of interactions between a pair of chromatin-sites; i.e., an enzyme like Topoisomerase-II binds to not all the pairs of beads simultaneously (let us say “global setting” here) but only selected pairs of beads transiently. We believe that the results from this setting is not the artefact of the model in the sense that some enzymes including Topoisomerase-II should work on pairs of chromatin-sites, according to the enzyme’s working mechanism, but not on a single site nor in a global setting.

Based on this consideration, we state that the genuine effect of Topoisomerase which we found is the microphase separation of the chromatic medium with emergent wall-like features (i.e., wall-like morphology and the negative nematic order therein). Indeed, we confirmed that these features stem from the transient nature of interactions between a pair of beads and has not been reported in earlier literature. In the subsection ‘Characteristic phase separation features, including wall-like organization of EC, due to Topo-II’ of the revised manuscript, we discuss how models with global settings fail to exhibit the characteristic features observed in our original model (MdDAM). Following that, we present simulation results suggesting that transient pair-catching mechanism of enzymatic activity is essential for the observed wall-like features.

The reviewer has also exemplified the concern that transiently switching off steric repulsion between the coarse-grained beads would alter osmotic pressure of the system, which may not be associated with Topo-II activity, and lead to the micro phase separation. Our brief answer to this specific concern is as follows: We agree that there should be reduction of osmotic pressure by the decrease of steric repulsion strength between AA pairs of beads. We saw, by the simulations of the model with transient removal of steric repulsion (“RNR” model in the revised manuscript), that indeed it can contribute to the microphase separation itself. However, such simulations did not give rise to the negative nematicity of AA bonds.

An additional point of discussion: We are expecting that such reduction of osmotic pressure is also partially a genuine effect, at least phenomenologically. We expect so because the affinity between two DNA segments associated with Topo-II function could affect the osmotic pressure. (In reality, the degree of reduction may not be large as given in our model, though.) Clarifying this point at a quantitative level could be an interesting future direction.

In that context there is a short sentence about effects from attraction due to topoisomerase activity: (l. 166) "the possibility … was ruled out". It is unclear to me how this can be "ruled out" and to me this sentence sounds too strong and not clear. A more careful discussion would help clarify these points.

We thank the editor and the reviewers for considering carefully and pointing this out. In our active polymer model, there exist a transient attraction state between two A beads caught by an enzyme. The idea behind including that short sentence in the previous version of our manuscript was to check if a non-transient effective attraction among the A beads can induce a phase separation like what we observe for some Λ>0 without any heterochromatin affinity. This was for a sanity check of our model and is not significant for the major findings of our research. So, in the revised manuscript, we have dropped that paragraph from the ‘Plausible mechanism …’ subsection and added the following sentence in the ‘Characteristic phase separation features, …’ subsection:

“We also considered a non-transient effective attraction model that may mimic the transient attraction state between A beads bound to an enzyme in MdDAM and found no phase separation at all for the case comparable to Λ=0.0309 (Figure 3—figure supplement 4)”.

Also, note that this revised figure supplement is for an effective attraction strength 0.24 s.u., whereas earlier it was mistakenly written as 0.25 s.u.

– I am somewhat puzzled by the HC affinity potential described in line 389. It vanishes for r=0 and for large r, so I cannot see that it describes an affinity. What is the physical meaning of epsilon_HC_ and why this choice of potential? In contrast the attractive potential due to enzyme activity is clearly an attractive potential. Why the different choices of potential?

Let us first focus on our model system for the monodisperse case without any enzymatic activity. In that case, a pair of B beads connected along the polymer have steric repulsion (hvex>0) among themselves. To set the bead sizes (say, dB) in our simulation setting, we set a criterion that the minimum of the combination (hspring+hvex) appears for a separation r=dB. We choose the heterochromatin affinity potential (hHC) acting on BB pairs in such a way that its minimum also appears at r=dB. In this way hHC does not disturb our chosen criterion to fix bead sizes. The parameter ϵHC in the heterochromatin affinity has dimension of spring constant. Please refer to Author response image 1 and the revised ‘Brownian dynamics’ description in the manuscript for further details.

Let us then discuss the case when a pair of A beads are caught by Topo-II. In that case, there is an absence of steric repulsion between them. So, to model the attraction between the caught A beads, we use an attractive Gaussian potential with its minimum appearing at r=0.

**Author response image 1. sa2fig1:** 

– The quantification of nematic bond order to demonstrate the wall-like nature of microphases is very interesting. However only the nematic order of A-A bonds is discussed. It would be good to show that the B-B bonds do not exhibit similar nematic order.

Interestingly, we note that enzymatic activity breaks local anisotropy of both the AA and the BB types of bonds. While for AA, we see negative nematicity, we see positive nematicity for BB bonds. Although one may doubt about the small magnitude of the order parameters, we must compare the cases of with and without enzymatic activity. We do not see nematic order without the activity, no matter if the system is phase separated or not (due heterochromatin affinity, see Figure 3—figure supplement 2). We appropriately discuss this observation in the revised manuscript.

Reviewer #2 (Recommendations for the authors):This computational work provides a new role of enzymatic activity in chromatin organization, especially Topoisomerase-II (Topo-II). The authors newly introduced a catch-and-release mechanism among euchromatin regions mimicking Topo-II activity and performed simulations of the polymer model. They show that the enzymatic activity promotes the microphase separation of the chromatin model. The model configurations seem consistent with the experimentally observed distribution of euchromatin and heterochromatin. Besides, they provide a theoretical framework for understanding the physical origin of the microphase separation using a simplified mean-field model. The mean-field calculation explains an effective attraction among heterochromatin due to the phantom and self-avoiding contributions, promoting a phase separation. The simulated configurations reveal a characteristic structure called wall-like organization of euchromatin components, which the mean-field framework cannot explain. These data suggest a possibility of forming a wall-like microphase separation in the cell nucleus by enzymatic activity.1) As polymer modeling approaches have revealed a phase-separated organization such as A/B compartments in the cell nucleus, the existence or assumption of the two type interactions on the active/inactive genomic regions should be a critical factor. This work assumes the catch-and-release mechanism among AA pairs and the attractive interaction among BB pairs. Therefore, the microphase separation would be predictable. However, the wall-like organization is not trivial and might become a universal phase-separated structure in a micro-scale.

We appreciate the reviewer’s effort to clarify the significance/non-triviality of this work. We agree with the reviewer on the above observation and would like to emphasize on the fact that the non-globule wall-like appearance of the A-domains with the associated negative nematic ordering of the AA bonds is a non-trivial finding of our study. We also note that in the presence of the enzymatic activity, the BB bonds also break the local isotropy and shows *positive* nematic ordering. In contrast to these observations, we do not see nematic ordering of AA and BB bonds for Λ=0, even though the system may be phase separated by heterochromatin affinity.

The outer walls in Figures 1d and 2b seem to be spherical and can be an effect of the spherical boundary condition. The authors do not address the possibility.

We have checked that the spherical appearance of the outer wall is due to the choice of a spherical geometry for the cavity. In the revised manuscript, we have included results for a sample simulation of our active polymer model within a cubic box with closed boundary in one direction (see Figure 3—figure supplement 3). We indeed see two planar walls appearing near the two closed boundaries of the box. Note that the characteristic phase separation features mentioned for the spherical cavity case survive in the cubic simulation box.

2) The reason why the authors change the volume fraction of A and fix the heterochromatin affinity as ε=4 in Figure 3 would be needed to clarify motivation in section "Wall-like organization of EC due to Topo-II."

During our research, we studied our model for different values of ϵHC and noticed that it phase separates into A/B for ϵHC=4 for all the volume fractions (ϕA) studied. Therefore, in Figure 3 we present data for this value of ϵHC so that we can compare the effect of enzymatic activity on phase configurations (viz., HC foci features) with and without it. Note that we tested the robustness of the obtained features for wider varieties of the values of ϵHC and ϕA (ϵHC
*= 0, 2, 4, 6,*
ϕA*=0.35, 0.40, 0.50, 0.60, 0.70*).

Altogether, in the corresponding section of the revised manuscript, we have added the following sentences:

“This is a robust feature we note in all of our simulations for different ϕA (0.35, 0.40, 0.50, 0.60, and 0.70) and ϵHC (0, 2, 4, and 6) in the presence of enzymatic activity. Figure 3 exemplifies the results for ϕA = 0.40 − 0.60 and ϵHC = 4, and also demonstrates a phase separation without enzymatic activity.”

3) Figure 3a shows the conversion of the A/B compartment configuration due to the enzymatic activity. Then, the authors characterize the wall-like organization of euchromatin by the local nematic order in Figure 3c. How about the local nematic order of BB bonds? The difference would strengthen the wall feature of A compartment regions.

As discussed above, we have seen positive nematicity of the BB bonds for Λ>0, and incorporated this observation in the revised manuscript. However, we keep the supporting plots as supplemental images and present the nematic ordering of the AA bonds only in Figure 3, because we find the non-globule wall-like features of A more surprising.